# Dopamine Receptor DRD2 Gene rs1076560, Personality Traits and Anxiety in the Polysubstance Use Disorder

**DOI:** 10.3390/brainsci10050262

**Published:** 2020-04-30

**Authors:** Aleksandra Suchanecka, Jolanta Chmielowiec, Krzysztof Chmielowiec, Jolanta Masiak, Olimpia Sipak-Szmigiel, Mariusz Sznabowicz, Wojciech Czarny, Monika Michałowska-Sawczyn, Grzegorz Trybek, Anna Grzywacz

**Affiliations:** 1Independent Laboratory of Health Promotion of the Pomeranian Medical University in Szczecin, 11 Chlapowskiego St., 70-204 Szczecin, Poland; o.suchanecka@gmail.com; 2Department of Hygiene and Epidemiology, Collegium Medicum, University of Zielona Góra, Zyty 28 St., 65-046 Zielona Gora, Poland; chmiele1@o2.pl (J.C.); chmiele@vp.pl (K.C.); 3Neurophysiological Independent Unit, Department of Psychiatry, Medical University of Lublin, 20-093 Lublin, Poland; jolantamasiak@wp.pl; 4Department of Obstetrics and Pathology of Pregnancy, Pomeranian Medical University, 48 Żołnierska St., 71-210 Szczecin, Poland; olimpiasipak-szmigiel@wp.pl; 5Indywidual Medical Practice MD M Sznabowicz, Lutówko 14, 74-320 Barlinek, Poland; moris500@wp.pl; 6Faculty of Physical Education, University of Rzeszów, Towarnickiego 3 St., 35-959 Rzeszów, Poland; wojciechczarny@wp.pl; 7Faculty of Physical Culture, Gdańsk University of Physical Education and Sport, Kazimierza Górskiego 1 St., 80-336 Gdańsk, Poland; monikamichalowska@op.pl; 8Department of Oral Surgery, Pomeranian Medical University in Szczecin, 72 Powstańców Wlkp. St., 70-111 Szczecin, Poland; g.trybek@gmail.com

**Keywords:** addiction, dopamine, dependence, personality traits, genetics

## Abstract

Development of an addiction is conditioned by many factors. The dopaminergic system has been shown to be the key element in this process. In this paper, we analyzed the influence of dopamine receptor 2 polymorphism rs1076560 in two groups—polysubstance-dependent male patients (n = 299) and the controls matched for age (n = 301). In both groups, we applied the same questionnaires for testing—Mini-international neuropsychiatric interview, the NEO Five-Factor Inventory, and the State–Trait Anxiety Inventory. The real-time PCR method was used for genotyping. When we compared the controls with the case group subjects, we observed significantly higher scores in the second group on both the state and trait scales of anxiety, as well as on the Neuroticism and Openness scales of the NEO-FFI; and lower scores on the scales of Extraversion and Agreeability of the NEO-FFI. The model 2 × 3 factorial ANOVA of the addicted subjects and controls was performed, and the DRD2 rs1076560 variant interaction was found for the anxiety state and trait scales, and for the NEO-FFI Neuroticism scale. The observed associations allow noticing that analysis of psychological factors in combination with genetic data opens new possibilities in addiction research.

## 1. Introduction

Some of the greatest health problems concerned globally are those connected with illicit substance use and abuse. The World Drug Report showed that in 2017, about 271 million people aged 15–64 (5.5% of the global population) had some experience with drug usage in the previous year. The data indicates that 585 thousand people lost their life because of drugs, and 42 million years of life in health were lost as an effect of drug use disorders [1]. Dependence is a result of complex interaction of genetic, epigenetic, and environmental factors. It is observed among predisposed individuals as an effect of their repetitive exposure to addictive chemicals and involves behavioral changes [2,3].

Addiction development is conditioned by many factors. The dopaminergic system showed to be the key element, with special attention to the mesolimbic dopamine system, which is an essential element of the reward system. It can be observed that substance use disorder and dependence may relate to changes in this system [4]. Shaping and maintaining both the control of emotional behavior and processing of emotions seemed to be directly conditioned by dopaminergic transmission [5]. One of the dopamine receptors, D2, is active in this process [6]. The analysis of personality and behavioral traits together with genetic variations in a group of substance-dependent individuals is a fairly novel way of deciphering the impact of given factors on dependence [7,8].

Human lifestyle and behavior are related with personality, the factor that is also connected with maintaining proper functions during a lifetime. The five-factor model, the so called Big Five, is one of the most used models in personality research [9,10,11,12]. The five-factor model consists of five traits: Openness, Conscientiousness, Extraversion, Agreeableness, and Neuroticism. All the abovementioned traits are associated with cognition, motivation, emotions, and behavior, and have impact on differences among individuals [13]. The revised NEO Five-Factor inventory (NEO-FFI) is a tool which is broadly used in analyses of these personality traits [10]. In his psychobiological theory of temperament and character, Cloninger indicated that extraversion is associated with dopaminergic pathways [14]. The same association was noticed by Costa and McCrea, and it is the essential element of their personality concept [15]. The State–Trait Anxiety Inventory (STAI) is a tool applied in measuring the state of anxiety (A-State), including fear, discomfort, and arousal of the autonomic nervous system occurring temporarily in relation to a particular situation, as well as the trait of anxiety (A-Trait), which can be described as a permanent and enduring disposition to have stress, worries, and discomfort [16].

The dopamine receptor D2 gene (DRD2) is located in chromosome 11q23 and spans 65,56 kilobase. The DRD2 gene includes 8 exons that undergo transcription to messenger RNA of 2713 kb that is translated to protein of 443 amino acids (AA). Skipping of the sixth exon leads to production of a short form of receptor in opposition to a 29 AA longer form of the receptor protein. The two isoforms of the D2 receptors differ in their affinities for inhibitory G-proteins [17]. The D2 receptors mediate the rewarding feelings of addictive substances. Substance use disorders, dependence and its endophenotypes are conditioned by polymorphisms occurring in the DRD2 gene and D2 receptor [18,19,20,21]. DRD2 gene rs1076560 located in its 6^th^ intron and its polymorphic versions are considered to be elements that can be analyzed in relation to genetics of behavior and psychiatry. The occurrence of the A allele of rs1076560 is connected with a lower expression of the short isoform (D2S) relative to the long one (D2L) in the caudate putamen and prefrontal cortex [22]. Multimodal imaging studies show that this polymorphism determines the D2 ligands binding in the striatum and reflects a correlation between the striatal D2 signaling and the activity of working memory in the prefrontal cortex. The carriers of the A allele have a reduced D2 receptor ligands binding [23]. The presence of the C/C genotype is associated with higher expression levels of the D2S mRNA in the striatum and prefrontal cortex as well as with better cognitive processing than the A/C genotype [24].

The role of dopamine, and especially the isoforms of the D2 receptor as an integrator of motivation, action, and emotion, is well documented in animal studies. Analysis of knock-out animals showed that D2L is the most important factor in mediating response to novel stimuli. The knock-out of D2L reduced explorative behavior and increased the time needed to escape from a harmful situation [25]. Another study showed that reduced availability of D2 receptors in the ventral tegmental area (VTA) heightened motivation for drug and food rewards [26].

In human studies, low avidity of D2 receptors is observed in individuals with alcohol dependence [27], although its higher level could protect against alcoholism [28]. The studies also indicate the association between the short isoform occurrence and schizophrenia [29]; what is observed is the fact that healthy carriers of the C/A genotype are more prone to higher levels of schizotypy than those with the C/C genotype [30,31]. The A allele is also associated with alcohol dependence [31], cocaine [22] and opiates [32] abuse. Furthermore, low D2 and D3 receptor binding and low presynaptic dopamine can influence the predisposition to heroin addiction [33]. Interestingly, alcohol elevates the D2L receptor level and decreases the D2S receptor percentage, both in vitro and in vivo, and reduces the dopaminergic agent’s inhibition of prolactin release. This may be considered an immediate impact on alternative splicing through mRNA [34]. Decreased somatodendritic D2 receptor availability influenced novelty-seeking and impulsiveness [35]. Decreased expression of these receptors in the hippocampus and nucleus accumbens was correlated with greater seeking behavior for the drug when its non-availability was signaled [36].

In the present study, we wanted to concentrate mainly on the analysis of DRD2 rs1076560 polymorphism in the group of individuals with polysubstance use disorder as well as in the controls. The other element of our analysis consisted of personality traits. Hence, we combined the analysis of influence of genetic factors with assessment of the personality traits that were measured with the NEO Five-Factor Personality Inventory and the State–Trait Anxiety Inventory Scale.

## 2. Materials and Methods

### 2.1. Participants

The study was held in the Independent Laboratory of Health Promotion, Pomeranian Medical University in Szczecin. The approval of the Bioethics Committee of the University (KB-0012/106/16) and an informed written consent of the participants were obtained. The study group consisted of male volunteers: polysubstance-dependent patients (n = 299; mean age = 28, SD = 6.45) and healthy, non-addicted male control subjects (n = 301; mean age = 22; SD = 4.57). The dependent patients were recruited at addiction treatment facilities in the province of Lubusz. The dependent patients were recruited after at least 3 months of abstinence; none of them were undergoing pharmacotherapy. Both groups were tested by the psychiatrist using the following questionnaires: the Mini-international neuropsychiatric interview (MINI), the NEO Five-Factor Inventory (NEO-FFI), and the State–Trait Anxiety Inventory (STAI). The MINI questionnaire was used to exclude subjects with the neuropsychiatric disorders other than dependence, as well as the subjects with any of these disorders from the control group. The history of substance dependence was obtained using the Polish version of ICD-10, medical history, and the authors’ survey.

### 2.2. Genotyping

DNA was obtained from the whole venous blood. The genomic DNA was isolated according to standard procedures. Samples were genotyped by the real-time PCR method using LightCycler 480 II (Roche Diagnostic, Basel, Switzerland). Genotyping was performed with the fluorescence resonance energy transfer using the real-time PCR method on the LightCycler^®^ 480 II System (Roche Diagnostic, Basel, Switzerland) according to the manufacturer’s protocol. The fluorescence signal was plotted against temperature to give melting curves for each sample. Peaks were obtained at 57.36 °C for the A allele and at 64.40 °C for the C allele.

### 2.3. Statistical Analysis

The DRD2 rs1076560 genotype distribution was tested according to the Hardy–Weinberg equilibrium (HWE) with the HWE software (http://www.oege.org/software/hwe-mr-calc.html).

The analyzed variables did not have a normal distribution. To determine the difference in the analyzed traits of the NEO Five-Factor Inventory (Neuroticism, Extraversion, Openness, Agreeability, Conscientiousness) and subscales of Anxiety (state and trait), the Mann–Whitney *U*-test was used.

Not all the assumptions required for the ANOVA analysis have been met. The assumption about the normal distribution was not fulfilled for all dependent variables, but the variance was the same (Levene’s test’s *p* > 0.05). As the number of subjects in groups was also large, it was therefore decided to use a multivariate 2 × 3 factorial ANOVA. The test was used to show an association between personality traits and anxiety subscales (STAI, NEO Five-Factor Inventory) and the addiction factor or its absence and the DRD2 rs1076560 polymorphism (personality traits × control and addicted subjects × genetic feature).

The frequencies of genotypes and alleles of the DRD2 rs1076560 polymorphism in the analyzed groups were compared by the chi-squared test; with the Bonferroni multiple comparisons correction, the accepted level of significance was 0.0071 (0.05/7). All the analyses were performed using STATISTICA 13 (Tibco Software Inc, Palo Alto, CA, USA) for Windows (Microsoft Corporation, Redmond, WA, USA).

## 3. Results

The frequency distributions accorded with the HWE. There was no statistical difference between the addicted subjects and the control subjects (Table 1).

The DRD2 rs1076560 genotypes and allele frequencies in the studied sample did not differ between the subjects of the analyzed groups (Table 2).

The means and standard deviations for all the NEO Five-Factor Inventory traits and STAI subscales in the group of addicted subjects and in the control subjects are presented in Table 3. In comparison with the controls, the case group subjects had significantly higher scores on the state (Mean (M) 5.90 vs. M 4.69, *p* < 0.0001) and trait (M 7.11 vs. M 5.16, *p* < 0.0001) scales of anxiety. The results were also higher for the Neuroticism (M 6.73 vs. M 4.67, *p* < 0.0001) and Openness (M 5.01 vs. M 4.53, *p* < 0.0071) traits measured by the NEO-FFI. Lower scores on the scales of Extraversion (M 5.76 vs. M 6.37, *p* < 0.001) and Agreeability (M 4.30 vs. M 5.60, *p* < 0.0001) were observed (Table 3).

### 3.1. STAI State Scale and DRD2 rs1076560

Results of the 2 × 3 factorial ANOVA of the DRD2 rs1076560 genotype were found for the STAI state scale (F_2594_ = 6.17, *p* < 0.0071, η^2^ = 0.020) (Table 4). Power calculation—our sample had more than 89% power to detect the DRD2 rs1076560 genotype’s main effects on the studied STAI state scale and their interaction effect (about 2% of the phenotype variance).

### 3.2. STAI Trait Scale and DRD2 rs1076560

Results of the 2 × 3 factorial ANOVA of the addicted subjects and the control subjects were found for the STAI trait scale (F_1593_ = 20.11, *p* < 0.0001, η^2^ = 0.033) and the DRD2 rs1076560 genotype was found for the STAI trait scale (F_2593_ = 9.06, *p* < 0.0071, η^2^ = 0.030) (Table 4). Power calculation—our sample had 99% power to detect effects of the studied STAI trait scale and their interaction effect (about 3% of the phenotype variance) and more than 97% power to detect the DRD2 rs1076560 genotype’s effects of the studied STAI trait scale and their interaction effect (about 3% of the phenotype variance) in the addicted subjects and the control subjects.

### 3.3. NEO Five-Factor Inventory Neuroticism Scale and DRD2 rs1076560

Results of the 2 × 3 factorial ANOVA of the addicted subjects and the control subjects were found for the NEO Five-Factor Inventory’s Neuroticism scale (F_1593_ = 26.02, *p* < 0.0001, η^2^ = 0.042) and the DRD2 rs1076560 genotype was found for the NEO Five-Factor Inventory’s Neuroticism scale (F_2593_ = 5.15, *p* < 0.0071, η^2^ = 0.017) (Table 4). Power calculation—our sample had 99% power to detect effects of the studied NEO Five-Factor Inventory’s Neuroticism scale and their interaction effect (about 4% of the phenotype variance) and more than 83% power to detect the DRD2 rs1076560 genotype’s effects of the studied NEO Five-Factor Inventory’s Neuroticism scale and their interaction effect (about 2% of the phenotype variance) in the addicted subjects and the control subjects. 

## 4. Discussion

In the present study, we concentrated on the relationship between DRD2 gene rs107656 and the personality traits measured with the Big Five Questionnaire (NEO-FFI), anxiety measured with the STAI to analyze the aspects modulating occurrence of substance dependence in the group of polysubstance-dependent patients. Dopamine is an essential factor in neural formation of a motivated behavior, and anomalies in the dopaminergic system seem to be essential elements in the etiology of motivation-related psychiatric disorders, mainly, dependence [37,38]. In our analysis, we observed that DRD2 rs1076560 variants influence the selected personality traits which are engaged in formation of a motivational behavior that can lead to dependence. Hence, both genetic factors and personality traits should be considered as the elements influencing vulnerability to developing dependence. We are still not sure if predisposing genetic variants and personality traits can be treated as a common or a specific factor influencing all classes of dependence. If we want to explain the relationship between genetic variances, personality traits, and their interaction in dependence, the analysis incorporating all these factors should be performed.

In our study, we did not observe presence of any association, neither genotypes nor alleles of rs1076560 with occurrence of polysubstance dependence. As far as we know, there exist no studies analyzing the rs1076560 polymorphism in the group of polysubstance-dependent subjects. Analysis preformed in groups with cocaine [22], opioid [32,39], and alcohol [31] dependence showed an association of substance dependence with this polymorphism, although the latter is not universally agreed on [40]. Our analysis indicates that the scores of the STAI inventory are significantly different for the cases and the controls. Dependent subjects had significantly higher scores for both anxiety trait and state scales. Moreover, results of the 2 × 3 factorial ANOVA indicated that rs1076560 was also significant for both scales of the STAI inventory. Anxiety disorders often co-exist with substance dependence and families with the substance use problem are more prone to its occurrence [41]. Increased values of anxious-impulsive personality traits are observed more frequently among individuals with substance use disorder and in their families than in controls. Therefore, anxious-impulsive personality traits can be considered possible endophenotypes increasing the risk of cocaine or amphetamine dependence development [42], and what is more, individuals with a higher level of anxiety are more susceptible to develop substance dependence [43]. What is important is the fact that different population studies seem to confirm the relationship between anxiety measured by the STAI and substance dependence [44,45]. A study from 2014 [45] noticed that the addicted group had increased scores not only in the STAI inventory, but the same situation was also observed in the depression scale, and a reverse one—in the stress tolerance scale. It is important to remember that both clinical and research data suggest that poor skills regarding coping with stress and negative mood states are a common motive for the use of psychoactive substances among heavy abusers [46]. We previously mentioned that rs1076560 is associated with the process of splicing. It is a prognostic factor of relative avoidance of the reward-seeking behavior; this results in the highest avoidance deficit in the individuals with the rs1076550 allele. As a result, carriers of the A allele were less anxious and prone to display worse avoidance. What is interesting in the association is the fact that these participants represent lowered expression of presynaptic D2 autoreceptors. Carriers of the rs1076560 C/C genotype displayed better avoidance learning, whereas carriers of the A variant showed lower avoidance but better reward learning [47]. Not only rs1076560 is associated with the D2 receptor function in the striatum. The other allele that influences D2 receptor availability in the striatum is located in the ANKK1 gene, and can be associated with the anxiety symptoms occurring from early childhood similarly to the A allele of rs1076560 [48]. The observation that both polymorphisms have, to some degree, alike effect and influence the feeling of anxiety strengthens the hypothesis of anxiety-related substance abuse.

Our NEO-FFI scores allow asserting existence of significant differences between the cases and the controls. The scores for the Neuroticism and Openness traits were increased and the scores for the Extraversion and Agreeability traits were decreased in the cases when compared to the controls. Moreover, results of the 2 × 3 factorial ANOVA indicated significant influence of rs1076560 on the Neuroticism trait. Most of the studies emphasize an important role of the trait of personality in the case of problematic substance use. Stress sensitivity, that is connected with neuroticism, can be treated as an element that differs in individuals with psychoactive substance abuse and their non-affected relatives; that allows thinking about it as an endophenotype of substance use disorder. An analysis of personality traits in addicted subjects [49] demonstrated that decreased scores of the Conscientiousness and increased scores of the Neuroticism traits are the factors influencing vulnerability to psychoactive substances usage, i.e., tobacco, heroin, and cocaine. Surprising is the fact that marijuana users are also low on the Conscientiousness scale, but at the same time, they are average on the Neuroticism and high on the Openness scales, unlike other substance users. It is worth mentioning that all the six facets of the Neuroticism trait were associated with tobacco, heroin, and cocaine use. The study of tobacco smokers [50] revealed that low scores on the Neuroticism and Openness scales were the factors conditioning tobacco abstinence, and high scores on the Neuroticism and low scores on the Agreeableness and Conscientiousness scales were associated with predictors of the worst outcomes, including a greater number of cigarettes smoked per day.

As far as we are concerned, our research is the first that in such a wide range analyzed the association of the splicing rs1076560 polymorphism and personality traits in substance-dependent subjects; however, we are also aware of its limitations. In our research, only males of Caucasian origin were studied; hence, we are aware of a need to replicate it in a female group, as well as in other ethnic origins. Furthermore, the study was based on a self-selected (i.e., non-representative) sample of male participants. The cross-sectional analysis does not allow drawing conclusions on causality.

## 5. Conclusions

Conclusively, the comparison of the controls with the case group subjects indicates that the first one represented significantly higher scores on both the state and trait scales of the STAI, as well as on the Neuroticism and Openness scales of the NEO-FFI, and lower scores on the scales of Extraversion and Agreeability of the NEO-FFI. Results of the 2 × 3 factorial ANOVA of the addicted subjects and the control subjects and the DRD2 rs1076560 variant interaction were found for the STAI state and trait scales and the NEO-FFI Neuroticism scale. The associations allow concluding that the combination of psychological factors and genetic data creates a new area of research in addiction studies. In order to better examine substance dependence, our future studies concerning this subject will incorporate not only the impact of genetic polymorphisms and personality traits, but will also add epigenetic analysis to this multivariable problem.

## Figures and Tables

**Table 1 brainsci-10-00262-t001:** Hardy-Weinberg equilibrium of DRD2 rs1076560 in the group of addicted subjects and controls.

Group	DRD2 rs1076560				
		Observed (Expected)	Alleles Frequency	χ^2^	*p*
Addicted subjects N = 299	C/C	197 (195.87)	p allele freq (C) = 0.81 q allele freq (A) = 0.19	0.18	>0.05
	A/C	90 (92.27)			
	A/A	12 (10.87)			
Controls N = 301	C/C	208 (205.98)	p allele freq (C) = 0.83 q allele freq (A) = 0.17	0.66	>0.05
	A/C	82 (86.03)			
	A/A	11 (8.98)			

N—number of subjects, *p*—statistical significance, χ^2^—chi-squared test result, freq—frequency.

**Table 2 brainsci-10-00262-t002:** Frequency of genotypes and alleles of the DRD2 gene rs1076560 polymorphism in the addicted subjects and in the controls.

Group	DRD2 rs1076560				
	Genotypes			Alleles	
	C/C N(%)	A/C N(%)	A/A N(%)	C N(%)	A N(%)
Addicted subjects	197	90	12	484	114
N = 299	(0.66)	(0.30)	(0.04)	(0.81)	(0.19)
Controls	208	82	11	498	104
N = 301	(0.69)	(0.27)	(0.04)	(0.83)	(0.17)
χ^2^	0.615			0.640	
*p*	0.735			0.422	

N—number of subjects, *p*—statistical significance, χ^2^—chi-squared test result.

**Table 3 brainsci-10-00262-t003:** Analysis of the State–Trait Anxiety Inventory, NEO Five-Factor Inventory results in the addicted subjects and in the controls.

STAI/NEO-FFI (sten)	Addicted Subjects (N = 299), M ± SD	Controls (N = 301), M ± SD	Mann–Whitney *U*-Test, *z*-Value	*p*
STAI state scale	5.90 ± 2.42	4.69 ± 2.14	**6.39**	**<0.0001**
STAI trait scale	7.11 ± 2.28	5.16 ± 2.18	**9.62**	**<0.0001**
NEO-FFI Neuroticism scale	6.73 ± 2.18	4.67 ± 2.01	**10.78**	**<0.0001**
NEO-FFI Extraversion scale	5.76 ± 2.14	6.37 ± 1.98	**−3.47**	**<0.001**
NEO-FFI Openness scale	5.01 ± 2.02	4.53 ± 1.61	**2.91**	**<0.0071**
NEO-FFI Agreeability scale	4.30 ± 1.93	5.60 ± 2.09	**−7.52**	**<0.0001**
NEO-FFI Conscientiousness scale	5.59 ± 2.27	6.08 ± 2.15	−2.62	>0.0071

The Bonferroni correction was used, and the *p*-value was reduced to 0.0071 (*p* = 0.05/7 (the number of statistical tests conducted)). M—mean, SD—standard deviation, Mann–Whitney *U*-test (*z*-value). STAI—State—trait anxiety Inventory; NEO-FFI—NEO Five Factor Inventory; Statistically significant between-group differences are marked in bold print.

**Table 4 brainsci-10-00262-t004:** Results of the 2 × 3 factorial ANOVA for the addicted subjects and the controls, incorporating the State–Trait Anxiety Inventory and the NEO Five-Factor Inventory results and DRD2 rs1076560.

			DRD2 rs1076560			Multifactorial ANOVA			
STAI/NEO-FFI (sten)	Addicted Subjects (N = 299), M ± SD	Controls (N = 301), M ± SD	C/C (N = 405), M ± SD	A/C (N = 172), M ± SD	A/A (N = 23), M ± SD	Factor	F (*p*-value)	ɳ^2^	Power (Alfa = 0.05)
STAI state scale	5.90 ± 2.42	4.69 ± 2.14	5.28 ± 2.39	5.09 ± 2.27	6.83 ± 2.10	intercept	**F_1593_ = 1119.46 (*p* < 0.0001)**	**0.654**	**1.000**
						addicted/control	F_1593_ = 3.282 (*p* > 0.05)	0.006	0.440
						DRD2 rs1076560	**F_2593_ = 6.17 (*p* < 0.0071)**	**0.020**	**0.891**
						addicted/control × DRD2 rs1076560	F_2593_ = 2.37 (*p* > 0.05)	0.007	0.479
STAI trait scale	7.11 ± 2.28	5.16 ± 2.18	6.15 ± 2.46	5.85 ± 2.33	7.87 ± 1.98	intercept	**F_1593_ = 1571.65 (*p* < 0.0001)**	**0.727**	**1.000**
						addicted/control	**F_1593_ = 20.11 (*p* < 0.0001)**	**0.033**	**0.994**
						DRD2 rs1076560	**F_2593_ = 9.06 (*p* < 0.0071)**	**0.030**	**0.975**
						addicted/control × DRD2 rs1076560	F_2593_ = 1.41 (*p* > 0.05)	0.005	0.302
NEO-FFI Neuroticism scale	6.73 ± 2.18	4.67 ± 2.01	5.69 ± 2.40	5.54 ± 2.19	7.00 ± 1.88	intercept	**F_1593_ = 1425.78 (*p* < 0.0001)**	**0.706**	**1.000**
						addicted/control	**F_1593_ = 26.02 (*p* < 0.0001)**	**0.042**	**0.999**
						DRD2 rs1076560	**F_2593_ = 5.15 (*p* < 0.0071)**	**0.017**	**0.830**
						addicted/control × DRD2 rs1076560	F_2593_ = 1.85 (*p* > 0.05)	0.006	0.386
NEO-FFI Extraversion scale	5.76 ± 2.14	6.37 ± 1.98	6.01 ± 2.09	6.18 ± 2.09	6.18 ± 1.79	intercept	**F_1593_ = 1482.47 (*p* < 0.0001)**	**0.714**	**1.000**
						addicted/control	F_1593_ = 1.04 (*p* > 0.05)	0.002	0.176
						DRD2 rs1076560	F_2593_ = 0.57 (*p* > 0.05)	0.002	0.144
						addicted/control × DRD2 rs1076560	F_2593_ = 0.67 (*p* > 0.05)	0.002	0.162
NEO-FFI Openness scale	5.01 ± 2.02	4.53 ± 1.61	4. ± 1.88	4.78 ± 1.76	5.04 ± 1.86	intercept	**F_1593_ = 1178.04 (*p* < 0.0001)**	**0.665**	**1.000**
						addicted/control	**F_1593_ = 4.96 (*p* > 0.0071)**	**0.008**	**0.605**
						DRD2 rs1076560	F_1593_ = 0.26 (*p* > 0.05)	0.001	0.091
						addicted/control × DRD2 rs1076560	F_2593_ = 0.25 (*p* > 0.05)	0.001	0.089
NEO-FFI Agreeability scale	4.30 ± 1.93	5.60 ± 2.09	4.94 ± 2.16	4.97 ± 2.03	5.00 ± 1.83	intercept	**F_1593_ = 1019.84 (*p* < 0.0001)**	**0.632**	**1.000**
						addicted/control	**F_1593_= 14.03 (*p* < 0.0071)**	**0.023**	**0.962**
						DRD2 rs1076560	F_2__593_ = **0**.10 (*p* > 0.05)	0.0003	0.065
						addicted/control × DRD2 rs1076560	F_2593_ = **0**.15 (*p* > 0.05)	0.0004	0.072
NEO-FFI Conscientiousness scale	5.59 ± 2.27	6.08 ± 2.15	5.88 ± 2.23	5.85 ± 2.18	4.86 ± 2.40	intercept	**F_1593_ = 1055.62 (*p* < 0.0001)**	**0.640**	**1000**
						addicted/control	F_1593_ = 0.18 (*p* > 0.05)	0.0003	0.071
						DRD2 rs1076560	F_2593_ = 2.22 (*p* > 0.05)	0.007	0.453
						addicted/control × DRD2 rs1076560	F_2593_ = 1.56 (*p* > 0.05)	0.005	0.331

The Bonferroni correction was used, and the *p*-value was reduced to 0.0071 (*p* = 0.05/7 (the number of statistical tests conducted)). M—mean, SD—standard deviation, Mann–Whitney *U*-test (*z*-value). Statistically significant between-group differences are marked in bold print.

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
