# Peer review of "Dopamine Receptor DRD2 Gene rs1076560, Personality Traits and Anxiety in the Polysubstance Use Disorder"

_brainsci, 2020, doi:10.3390/brainsci10050262_

Round 1
Reviewer 1 Report
In their manuscript “Dopamine receptor DRD2 gene rs1076560, personality traits and anxiety in the polysubstance use disorder”, Suchanecka and colleagues assess how DRD2 gene variants influence personality and emotions in individuals with polysubstance dependence disorder. The authors demonstrate that DRD2 gene rs1076560 is associated with vulnerability to develop anxiety and personality alternations, indicating a gene-environment interaction. Overall the manuscript is clearly written and topically relevant. I feel that the manuscript is a strong candidate for publication in Brain Sciences, but have a number of minor comments that should be addressed prior to publication.
Comments:
(1) The experimental designs and observations in this manuscript are complementary and relevant to a recent study published by the same group (Grzywacz et al., 2020, Int J Environ Res Public Health.). Therefore, correcting the genotypes of rs1076560 to C/C, A/C, and A/A as presented previously would be helpful.
(2) The MMSE score is not presented in the current manuscript as stated in the abstract.
(3) Line 206, “In the present study we concentrated on relation between DRD2 gene rs107656 polymorphism”. It should be rs1076560.
Author Response
Dear Reviewer,
We would like to thank you for your valuable comments on the article. Below you will find our reply to your review. All changes are with a description or a comment and changes have been made to the manuscript (track changes in the tracking group on the review tab).
Reviewer 1
Comments and Suggestions for Authors
In their manuscript “Dopamine receptor DRD2 gene rs1076560, personality traits and anxiety in the polysubstance use disorder”, Suchanecka and colleagues assess how DRD2 gene variants influence personality and emotions in individuals with polysubstance dependence disorder. The authors demonstrate that DRD2 gene rs1076560 is associated with vulnerability to develop anxiety and personality alternations, indicating a gene-environment interaction. Overall the manuscript is clearly written and topically relevant. I feel that the manuscript is a strong candidate for publication in Brain Sciences but have a number of minor comments that should be addressed prior to publication.
Comments:
(1) The experimental designs and observations in this manuscript are complementary and relevant to a recent study published by the same group (Grzywacz et al., 2020, Int J Environ Res Public Health.). Therefore, correcting the genotypes of rs1076560 to C/C, A/C, and A/A as presented previously would be helpful.
Thank you for this valuable suggestion – the results, and information about associations of analysed polymorphism presented in introduction and discussion were changed as well to ensure comfort for the Readers.
(2) The MMSE score is not presented in the current manuscript as stated in the abstract.
Thank you very much for this incredibly important catch, due to an editing error on our site wrong name of test was put in the manuscript. The correct name of the test is Mini-International Neuropsychiatric Interview (M.I.N.I.). It was used in order to exclude subjects with other than substance dependence diagnosis from the case group, and to exclude subjects with any neuropsychiatric disorders from control group. Hence, the scores were not presented in the manuscript. The changes were made in lines 31 and 122 – 125. Thank you very much once more for this question. It saved us immensely.
(3) Line 206, “In the present study we concentrated on relation between DRD2 gene rs107656 polymorphism”. It should be rs1076560.
Thank you for this catch – the word “polymorphism” was deleted – line 210.
Reviewer 2 Report
Thank you for providing me the opportunity to review this paper regarding dopamine receptors, personality and substance abuse. Overall the paper is interesting, but there are a few issues that might be clarified.
Please explain why the study was limited to male participants.
What substances were included in the notion of ‘polysubstance’. Does it include substances that are freely available such as over-the-counter medication, tabaco, alcohol; prescription medication; illicit or street drugs?
How were control participants recruited?
At the end of the discussion, there could be a few sentences regarding study limitations. For example, the study was based on a self-selected (i.e. non-representative) sample of male participants. The cross-sectional analysis does not allow to draw conclusions on causality.
Also, there may be a few implications for further research or clinical practice.
Author Response
Dear Reviewer,
We would like to thank you for your valuable comments on the article. Below you will find our reply to your review. All changes are with a description or a comment and changes have been made to the manuscript (track changes in the tracking group on the review tab).
Reviewer 2
Comments and Suggestions for Authors
Thank you for providing me the opportunity to review this paper regarding dopamine receptors, personality and substance abuse. Overall the paper is interesting, but there are a few issues that might be clarified.
Please explain why the study was limited to male participants.
Thank you for this comment – we analysed only male participants due to differences in psychology, genetics, and biochemistry of both sexes. Those differences and their impact on dependence are well described in the literature on substance dependence. Results of analysis performed on group with mixed sexes would be scientifically doubtful. Hence, we decided to recruit to our study only males in order to analyse one, large homogenous group of subjects. In the future we plan to conduct separate study concerning female participants.
What substances were included in the notion of ‘polysubstance’. Does it include substances that are freely available such as over-the-counter medication, tabaco, alcohol; prescription medication; illicit or street drugs?
In our study polysubstance dependence is described according to ICD-10 classification code F19.2.
How were control participants recruited?
Control subjects recruited to our study were volunteers who applied to be part of our study based on internet announcement.
At the end of the discussion, there could be a few sentences regarding study limitations. For example, the study was based on a self-selected (i.e. non-representative) sample of male participants. The cross-sectional analysis does not allow to draw conclusions on causality.
Thank you for this comment – suggested study limitations were added – lines 278-279.
Also, there may be a few implications for further research or clinical practice.
Thank you for this suggestion – implications for further research were added – lines 287-290.